# Neuronal Metabolism and Neuroprotection: Neuroprotective Effect of Fingolimod on Menadione-Induced Mitochondrial Damage

**DOI:** 10.3390/cells10010034

**Published:** 2020-12-29

**Authors:** Antonio Gil, Elisa Martín-Montañez, Nadia Valverde, Estrella Lara, Federica Boraldi, Silvia Claros, Silvana-Yanina Romero-Zerbo, Oscar Fernández, Jose Pavia, Maria Garcia-Fernandez

**Affiliations:** 1Department of Pharmacology and Pediatrics, Faculty of Medicine, Malaga University, 29010 Malaga, Spain; antoniogilcebrian@gmail.com (A.G.); emartinm@uma.es (E.M.-M.); oscar.fernandez.sspa@gmail.com (O.F.); 2Neuroscience Unit, Biomedical Research Institute of Malaga (IBIMA), Malaga University Hospital, 29010 Malaga, Spain; nadiavm@uma.es (N.V.); elara@uma.es (E.L.); silviacg@uma.es (S.C.); 3Department of Human Physiology, Faculty of Medicine, Malaga University, 29010 Malaga, Spain; yaninaromero@uma.com; 4Department of Life Sciences, University of Modena e Reggio Emilia, 41125 Modena, Italy; federica.boraldi@unimore.it

**Keywords:** sphingosine-1-phosphate receptor analogue, fingolimod phosphate, neuroprotection, mitochondrial damage, glycolytic pathway, pentose phosphate pathway, redox homeostasis

## Abstract

Imbalance in the oxidative status in neurons, along with mitochondrial damage, are common characteristics in some neurodegenerative diseases. The maintenance in energy production is crucial to face and recover from oxidative damage, and the preservation of different sources of energy production is essential to preserve neuronal function. Fingolimod phosphate is a drug with neuroprotective and antioxidant actions, used in the treatment of multiple sclerosis. This work was performed in a model of oxidative damage on neuronal cell cultures exposed to menadione in the presence or absence of fingolimod phosphate. We studied the mitochondrial function, antioxidant enzymes, protein nitrosylation, and several pathways related with glucose metabolism and glycolytic and pentose phosphate in neuronal cells cultures. Our results showed that menadione produces a decrease in mitochondrial function, an imbalance in antioxidant enzymes, and an increase in nitrosylated proteins with a decrease in glycolysis and glucose-6-phosphate dehydrogenase. All these effects were counteracted when fingolimod phosphate was present in the incubation media. These effects were mediated, at least in part, by the interaction of this drug with its specific S1P receptors. These actions would make this drug a potential tool in the treatment of neurodegenerative processes, either to slow progression or alleviate symptoms.

## 1. Introduction

Energy is a key factor in maintaining brain function, especially for the generation of action potentials, axonal transport, synthesis and release of neurotransmitters, and synaptic function. In brain, the main sources of energy are based in the uptake and metabolism of glucose and oxygen [1], and the choice between one of those may have important consequences for brain function in both health and disease [2]. Several studies have demonstrated that many neurodegenerative diseases are triggered or maintained by metabolic alterations [3,4]. During aging, a certain decrease in glucose and oxygen metabolism can be seen [3], but a more dramatical decrease is found in disorders such as Alzheimer’s disease (AD), amyotrophic lateral sclerosis (ALS), Parkinson’s disease (PD), and Huntington’s disease (HD) [5]. The need of energy to maintain its function and integrity is much more manifest in neurons susceptible to degeneration, and therefore the failure in energy metabolism, along with the enormous energy demand, can eventually result in higher cellular stress in these neurons [5].

Some neurodegenerative diseases, such as PD, multiple sclerosis (MS), ALS, or AD, are characterized by an imbalance in the oxidative status of the cells, leading to neuronal damage and death, contributing to the disease pathogenesis [6]. These pathologies have in common a mitochondrial damage that in turn increases, even more, the oxidative damage on the neurons in such way that they could be grouped as mitochondriopathies [7]. In these, and probably in most neurodegenerative diseases, the importance of glycolysis on synapse, axonal conduction, and plasticity among other neuronal functions has been previously established [8]. The coexistence of different sources of ATP production, such as mitochondrial and glycolytic ATP, allows faster adaptative mechanisms to situations of high energy demand and may help in the maintenance of neuronal function in stress situations [9]. A clearer knowledge of the influence of the oxidative damage on these pathways could be crucial in the development of potential therapeutic strategies targeting these diseases [10].

Fingolimod is one of the few drugs available orally for treatment of MS; after phosphorylation, it is converted into fingolimod phosphate (FP), the active part of the compound. Recent works have indicated a neuroprotective effect of this drug that could promote an improvement in cognitive function in ischemic processes [11] and neurodegenerative disorders such as HD [12] or AD [13].

Several external and/or internal noxious stimuli are able to produce neuronal damage that in most occasions triggers an imbalance in the neuronal oxidative homeostasis, resulting in most cases in accumulation of oxygen free radicals that damage proteins, nucleic acids, and lipid membranes, leading to neuronal death [10]. To study neuronal dysfunction, we used a model based in the increase of endogenous reactive oxygen species (ROS) production, induced by the treatment of cells with the toxin menadione (VitK3) [14,15,16]. VitK3 is an exogenic toxin that produces mitochondrial damage linked to an increase in ROS [17], distorting cellular structures such as the endoplasmic reticulum, but also influencing important metabolic pathways such as glycolysis and/or pentose phosphate (PPP) [18]. VitK3 interferes in the quinone redox-cycle, uncoupling the mitochondrial respiratory chain, triggering the release of O_2_^−•^ that in the presence of Fe from the respiratory complex produces an increase in free radicals [17], with an increase in respiration without the increase in ATP production. Furthermore, oxidative stress could also activate/inactivate other metabolic pathways that may contribute to VitK3 damage [19].

In the present work, we resorted to the analysis of glucose metabolism pathways (glycolysis and pentose phosphate (PPP)), along with mitochondrial respiration, in order to investigate the molecular events associated with the protective effect of fingolimod against VitK3-induced oxidative cell damage in SN4741 neuronal cells. The results of this work will provide an insight into the underlying mechanisms in oxidative damage and neuroprotection.

## 2. Materials and Methods

### 2.1. Cell Culture and Treatments

The SN4741 (RRID:CVCL_S466) dopaminergic cell line derived from mouse substantia nigra [20] were cultured in D-MEM high glucose supplemented with 10% FCS, penicillin–streptomycin, and L-glutamine (Fisher Scientific SL, Madrid, Spain), which were incubated at 37 °C in 5% CO_2_ up to about 70–80% confluence. Cells were seeded in a 100 mm^2^ dish (5 × 10^6^ cells) or glass bottom 35 mm^2^ dish and 6-well plates (2 × 10^5^ cells each) and treated with 5 μM of VitK3 (CAS nº 58-27-5; Sigma, Madrid, Spain) in the absence or presence of 50 nM FP; this concentration was established in a previous work of our group [17], based in experiments on the toxicity of the drug in a range from 0.1 to 100 nM on control cultures, and experiments where different concentrations of FP were tested for the ability to protect neuronal cultures against the damage produced by Vitk3, assessed by cell viability. In order to obtain a tighter control on the concentration of drug in the culture media, in this work, we used in all the incubations of the active metabolite FP kindly provided by Novartis Pharmaceutical, instead of the prodrug. The sphingosine-1-phosphate (S1P) receptor antagonist W123 (W) 10 μM (CAS nº 1345982-24-2. Cayman Chemicals, Michigan, USA) was also co-incubated with VitK3 and FP. The treatments were carried out in Locke’s solution (modified) (137 mM NaCl, 5 mM CaCl_2_, 10 mM KCl, 25 mM glucose, 10 mM HEPES, pH 7.4) supplemented with penicillin–streptomycin and L-glutamine during 2 h at 37 °C. In some experiments, such as extracellular acidification rate (ECAR) and oxygen consumption rate (OCR), the incubation time was extended over 4 additional hours. In ECAR experiments, dishes, plates, and coverslips were pre-coated with poly-D-lysine. The groups studied in this work were control cells without treatment (CO), cells incubated in the presence of VitK3 (VitK3), cells treated with VitK3 in the presence of FP (VitK3+FP), cells treated with VitK3 in the presence of FP and the antagonist of S1P receptor W (VitK3+FP+W), and cells incubated only in the presence of FP (FP). In some experiments, performed to assess the effect of FP in the recovery of VitK3 damage, after incubation of 2 h with VitK3, the buffer was changed by only buffer (in control cells) or buffer with 50 nM FP (treated cells) and the same but in the presence of 10 μM W to clarify the contribution of the S1P receptor on this recovery; measurements were obtained over 4 additional hours.

### 2.2. Antioxidant Enzyme Activity

Antioxidant enzyme activity of superoxide dismutase (SOD; E.C.1.15.1.1), glutathione peroxidase (E.C.1.11.1.9) (GPX), and catalase (E.C.1.11.1.6) (CAT) was measured spectrophotometrically using a Cobas Mira autoanalyzer (ABX Diagnostics, Montpellier, France) and the different commercial kits described previously [21].

### 2.3. Protein Nitrosylation

Protein nitrosylation was measured in a homogenate of neuronal cultures using the commercial kit OxiSelect Nitrotyrosine ELISA Kit (Cell Biolabs, Inc., San Diego, CA, USA) following the supplier instructions.

### 2.4. Determination of Aldolase

Aldolase activity (EC 4.1.2.13) in cell homogenates was assessed using an Aldolase assay kit (Spinreact SP, Gerona, Spain) adapted to a Cobas Mira Autoanalyzer. This assay was based on the observation of the decrease in absorbance at 340 nm caused by the conversion of NADH to NAD^+^ [22].

### 2.5. Glucose-6-Phosphate-Dehydrogenase Activity

Glucose-6-phosphate-dehydrogenase (EC 1.1.1.49) (G-6-PDH) activity in cell homogenates was assessed using a G-6-PDH assay kit (Spinreact SP, Gerona, Spain) adapted to a Cobas Mira Autoanalyzer. This assay was based on the observation of the decrease in absorbance at 340 nm caused by the conversion of NADPH to NADP^+^ [22].

### 2.6. Extracellular Acidification Rate (ECAR) and Mitochondrial Oxygen Consumption Rate (OCR)

Proton excretion (glycolysis) produces rapid changes in the concentration of free protons in culture media; this change was measured as ECAR, using microplates in a Seahorse Bioscience XF24 analyzer (Agilent Technologies. Agilent, CA, USA). OCR was measured using the same instrument [23,24]. Cells were seeded at 2 × 10^4^ cells per well for 18 h prior to the analysis, and each experimental condition was performed on 8 replicates. Before each measurement, the cells were washed with PBS, and 590 μL of Agilent Seahorse XF Base Medium (without phenol red and bicarbonate) supplemented with 1 mM pyruvate and 25 mM glucose was added to each well. Measurements were normalized according to protein concentration in each well. For OCR experiments, we used the commercial kit “Seahorse XF cell Mito Stress test kit” (Agilent Technologies. Agilent, CA, USA) following the manufacturer’s instructions; this kit uses oligomycin, carbonyl cyanide-4-(trifluoromethoxy)phenylhydrazone (FCCP), and rotenone/antimycin as mitochondrial toxins. For the study of different situations of ECAR we used to uncouple mitochondrial respiration from ATP synthesis, we used 2,4-dinitrophenol (2,4-DNP); 2-deoxy-D-glucose (2-DG) was used to inhibit glycolysis and rotenone was used to inhibit NADH hydrogenase/complex I [23]. In all the experiments, data were normalized with protein content in each well, as described previously [25].

### 2.7. Immunocytochemical Staining

Immunocytochemical staining was performed as described previously [17]. The primary antibody was Anti-S1P Receptor EDG1 (1:100 *v*/*v*) (cat nº SAB4500687, Sigma, Madrid, Spain) in PBS/3% BSA/0.02% sodium azide, incubated at 4 °C overnight. Secondary antibody was Alexafluor 488 (Fisher Scientific SL, Madrid, Spain), in PBS/BSA, incubated for 30 min at room temperature in the dark. Images were acquired using an Olympus BX51 epifluorescence microscope at 40× magnification and processed using ImageJ software (U.S. National Institute of Health; http://rsbweb.nih.gov/ij/).

### 2.8. Statistical Analysis

Statistical differences were determined using one-way ANOVA. Pairwise comparisons were performed using a post hoc Newman–Keuls multiple comparison test. Statistical significance was considered to be *p* < 0.05. For data in which the measured units were arbitrary, the respective values represent the percentage relative to the control value unless otherwise specified.

## 3. Results and Discussion

In our work, we resorted to the analysis of glycolytic pathway and PPP, along with mitochondrial function, in order to study the molecular events responsible of the protective effect of FP against VitK3-induced oxidative damage. This study was performed on neuronal cell cultures and thus one of the limitations of the work was our failure in knowing the protective effect of FP in the whole animal. In any case, a clearer knowledge of these pathways could help to understand the mechanisms involved in oxidative damage, common in several neurodegenerative diseases, hence allowing for amelioration of symptoms and/or development of new therapeutic strategies.

On the basis of the importance of glycolysis to maintain neuronal metabolism when mitochondrial dysfunction appears, along with previous work by our group and others showing the influence of ROS in energy production by neurons [17,26,27], we decided to go deeper into the study not only of antioxidant enzymes and mitochondrial function, but also the effect of FP on glucose metabolism.

### 3.1. Mitochondrial Function Studies

In order to determine the influence of oxidative radicals on mitochondrial respiration chain function, we studied the OCR in neuronal cultures after oxidative damage induced by VitK3 in the presence or absence of FP. VitK3 is a derivative of vitamin K and a redox cycling agent, and its mechanism of action is well characterized. VitK3 is reduced at the level of complex I of the mitochondrial respiratory chain, decreasing its activity by 50% [28] and inhibiting complex IV activity [17] with an increase in respiration without increase in ATP production and decrease of spare respiratory capacity (SRC) and maximal respiratory capacity. Bioenergetic assessment is shown in Figure 1; the OCR data obtained in these experiments enabled the determination of oxygen consumption due to (a) ATP synthesis, (b) mitochondrial oxygen reserve capacity, (c) related to proton leak, (d) SRC, and (e) maximal respiration.

In basal situations (without addition of oligomycin, FCCP, and rotenone/antimycin), the incubation of neurons with VitK3 showed an increase in OCR of 17% compared to the control, agreeing with our previous results and others [17,19], whereas neurons incubated with VitK3 in the presence of FP maintained the same levels as the control cells (Figure 1a).

Mitochondrial ATP-dependent OCR is the OCR required for the synthesis of ATP at complex V, which can be easily studied as it is sensitive to oligomycin. Regarding oxygen consumption by mitochondrial ATP synthesis, this decreases by 25% after incubation with VitK3, agreeing with the results of Lakhter et al. [29], who also found a similar decrease after incubation with this toxin. In our experiments, FP partially rescued the VitK3-induced loss of ATP linked to OCR, agreeing with Bai et al. [30], who found that increased ATP production by FP could be the reason for its neuroprotective effect; in our case, FP restored OCR linked to ATP production to levels close to control neurons (Figure 1b).

We did not find differences in proton leak (Figure 1c), although we could see a clear difference in SRC, the difference between OCR at basal and at maximal respiratory activity, after oligomycin and FCCP addition. In our experiments, incubation of neurons with VitK3 produced a great decrease in SRC (40%) compared to the control, which was reverted to values close to the control in the presence of FP (Figure 1d).

SRC is essential to maintain neuronal homeostasis against oxidative and/or other types of cellular stress [31]. The use of mitochondrial SRC by neurons is very variable, ranging from approximately 6 to 7% in resting situation to up to 80% in firing neurons; thus, deterioration in mitochondrial SRC can be fatal to neurons [31]. The ability to increase the SRC by FP will allow mitochondria to produce more ATP and overcome an increase in oxidative stress imbalance. FP could then be considered a long-term regulator of mitochondrial respiration through the changes promoted in SRC; this modulation could be related to an increase in the expression of neuronal nitric oxide synthase (NOS) and a consequent increase in nitric oxide (NO) levels [32], agreeing with our results on protein nitrosylation found in Table 1. These long-term regulators produce permanent changes in mitochondrial respiration that seem to be tissue-specific [33]. This would make FP a valuable therapeutic tool in neurodegenerative diseases, such as PD, HD, or MS, in which imbalance in oxidative stress would produce a dramatical mitochondrial affectation in such a way that they have been grouped as mitochondriopathies [7,34,35].

We also found a similar result in maximal respiratory activity, which is the maximum rate of respiration achieved by a neuron after addition of uncoupling FCCP and oligomycin and an indicator of potential mitochondrial dysfunction [31]. In these experiments, incubation of neurons with VitK3 produced a clear decrease (32%) in maximal respiratory activity. This decrease was counteracted by coincubation in the presence of FP, being able to recover 20% of maximal respiratory activity (Figure 1e).

The effect of FP on mitochondrial function could be related to the modulation of mitochondrial translocator protein expression (TSPO) [36], a mitochondrial membrane protein associated with the mitochondrial permeability pore that is expressed in all cells of embryonic origin [37], such as those used in this work; this has been seen in MS, where FP decreases the expression of TSPO [38], stabilizing mitochondria. Furthermore, we could not exclude the possibility that PF may regulate the expression of prohibitins, a chaperone that regulates the mitochondrial respiratory complex assembly and activity [39,40,41,42]. There are also some reports on the protective effect of the stimulation of S1P1 receptor on genes involved in the expression of proteins that participate in the stabilization of mitochondrial function [36]. We could also postulate that FP induces an increase of protective factors, such as the nuclear factor erythroid 2-related factor 2 (Nrf2), a transcription factor that binds to antioxidant response elements (ARE) in the nucleus, leading to transcription of ARE genes. FP also facilitates the Nrf2 translocation to the nucleus [17], increasing the synthesis of mitochondrial enzymes and modulating mitochondrial function; these effects may delay or ameliorate the damage induced by VitK3 [43]. In all the experiments shown in Figure 1, control cells were incubated in the presence of FP without any significative change in the values obtained compared to control cells.

### 3.2. Antioxidant Enzymes Studies

In the study of enzymes related with O_2_^−•^ metabolism, we found an increase (77.3%) of mitochondrial SOD2 activity in VitK3-treated neurons compared to the control (Table 1). In our experiments, that increase was not compensated by variations in CAT or GPX activities (see ratio in Table 1).

Increased SOD2 activity would in turn produce an increase in H_2_O_2_ that, in order to maintain homeostasis, should be followed by increases in CAT and/or GPX to remove H_2_O_2_ excess and avoid oxidative damage. This would suggests that mitochondria are involved in the generation of oxidative stress triggered by VitK3, agreeing with our previous results [17]. SOD2 is an inducible mitochondrial enzyme that promotes the formation of H_2_O_2_, with one of the factors that induces its expression being the increase in O_2_^−•^ [44]. Interestingly, as we found in the experiments, in the presence of FP, the SOD2 activity was normalized to control levels, indicating that FP, as shown above, exerts a protective effect on mitochondrial function; it also reduced O_2_^−•^ production, as demonstrated previously [17], and both in turn could contribute to the reduction in the induction of SOD2.

GPX is an enzyme family with peroxidase activity that protects cells from oxidative damage on the basis of its ability to reduce free H_2_O_2_ to water at low H_2_O_2_ concentrations. In our experiments, we found a decrease (79.4%) in GPX activity in neurons treated with VitK3 compared to the control, probably as a consequence of the increase in O_2_^−•^/SOD, decrease in total thiols, and increase in nitrosative stress [45], which exerts actions in parallel with ROS, as demonstrated previously [17]. In these experiments, the inclusion of FP in the incubation media partially recovered (50%) the activity of this enzyme, agreeing with the finding of other authors [46]; this effect could be mediated by the Nrf2 translocation to the nucleus, as demonstrated previously [17,47].

CAT is an enzyme that efficiently removes H_2_O_2_ at high concentrations of the oxidant. In our experiments, we did not see any modification after incubation with VitK3 alone or in presence of FP; only in control cells was a clear increase (57%) in CAT activity produced. This could be related with ability of FP to promote an increase in Nrf2 expression and translocation demonstrated in a previous work [17].

To assess the putative damage produced by oxidative stress induced by VitK3 treatment, we resorted to the study of protein nitrosylation. The nitrosylation of proteins triggered by NO is greatly enhanced in presence of O_2_^−•^ with the formation of peroxinitrites [48], and it is an important regulator of mitochondrial activity involved in energy-transducing systems [49]. NO is an important molecule in neuronal cell signaling, but its overproduction can be toxic to neurons, causing protein nitrosylation and mitochondrial damage [50]. In our experiments, we found a great increase (almost 600%) in protein nitrosylation in neurons treated with VitK3; again coincubation in presence of FP restored nitrosylation to levels close to control neurons, which could be explained by the ability of FP to inhibit the inducible NO demonstrated previously [12] that in turn would decrease protein nitrosylation. We also could see an increase in control neurons treated with FP (246%), which could have been due to the stimulation of neuronal NO [32]; moreover, FP has been involved in an increase in neuronal plasticity mediated by NO [51,52], although further studies will be necessary to clarify this point.

### 3.3. Glucose Metabolism Studies

In neuronal cells, glucose is mainly metabolized via glycolysis, the PPP, and oxidative phosphorylation to produce ATP. Expression of genes associated with plasticity and synaptic remodeling processes are associated in the adult brain with aerobic glycolysis [9]. Active synapses obtain its energy by glucose breakdown through glycolysis, with this being the main source of cellular energy, as well as the preferred method for certain neuronal functions, such as fast axonal transport [53].

In presence of VitK3, we found a great decrease (46% compared to control) in glycolysis, measured as ECAR after 2 h of incubation; coincubation in the presence of FP recovered values to control levels (Figure 2a,b). This effect was similar to that found with some sphingosine-1-phospate mimetics that increase the expression of genes related with glycolysis, ECAR, and glycolytic capacity [54]. In the study of glycolytic capacity, which measures the maximum rate of conversion of glucose to pyruvate or lactate, we found that VitK3 produced a decrease by 47% compared to the control, with this decrease being totally reverted by coincubation in the presence of FP (Figure 2a,c). When we studied the glycolytic reserve, which is the ability of cells to adapt to extra energetic demands when the glycolytic function is working at maximum, neurons incubated in presence of VitK3 also showed a reduction in glycolytic reserve by 48% compared to the control; again, coincubation in the presence of FP reverted this value to control levels (Figure 2a,d). Our results partially agree with those from Lee et al. [54], who found an increase of glycolytic capacity with S1P mimetics in mesenchymal cells, although the authors did not find differences in glycolytic reserve, possibly due to the different type of cells. This recovery in glycolytic capacity would rapidly supply ATP in situations where it is needed and when the mitochondrial metabolism, although generally having a high capacity to produce ATP, is not totally functional, as demonstrated previously.

As mentioned above, we found an increase in the expression of metabolic pathway related to glycolysis. This has also been described before in situations of energy stress, where increase in glycolytic metabolic enzymes have been seen in presynaptic buttons [55]. The decrease in glycolytic enzyme activity as a consequence of oxidative damage, seen in early stages of some neurodegenerative diseases, can lead to a decrease in ATP synthesis, which in turn would lead to a less reduced environment and an increase in free radical production [56]. In some neurodegenerative diseases, especially in those where axonal conduction is damaged, neurons tend to restore the conduction by increasing energy consumption to restore axonal function [57]. In AD, low glucose metabolism is associated with cognitive failure [58]; moreover, Rone et al. highlight the role of glycolysis to obtain energy in cells under metabolic stress [59]. One of the key enzymes involved in glycolysis is aldolase. In this work, we found a decrease (23%) in aldolase activity after treatment of neurons with VitK3. Again, FP restored values to those found in control neurons (Figure 3), agreeing with Lee et al., who also found an increase in aldolase activity with S1P mimetics [54]; moreover, Geffin et al. found an upregulation of genes involved in glycolysis, including aldolase, in cells exposed to HIV and treated with FP [60]. On the basis of these findings, FP could be considered as a new therapeutic approach to the treatment of neurodegenerative diseases showing a decrease in glycolysis, such as PD, AD, or MS [26].

Although our model was based on a mitochondrial oxidative damage, we saw a decrease in glycolytic function that was recovered in the presence of FP. Some authors have demonstrated that withdrawal of FP after a period of treatment could trigger relapses in MS patients [61], and thus we decided to maintain FP in one of the groups, extending the study time to determine the need of FP in the medium to maintain its beneficial effects. In these experiments, we found a decrease in ECAR compared to control neurons; FP almost totally reverted the damage produced by VitK3, thus maintaining glycolytic activity to levels close to control up to the maximum time studied, whereas in those neurons not in presence of FP, the damage was maintained over time (Figure 4). This could be of great interest, as many neurodegenerative diseases have in common a decrease in energy metabolism and hence in ATP levels [62]. In these situations, enhancement of glycolytic processes may represent a good approach to slow progression and/or alleviate symptomatology [62,63].

In the processing of glucose, the balance between glycolysis and PPP appears to be important, as the PPP processing of glucose serves as a source of NADPH that in turn will regenerate oxidized antioxidants such as glutathione and thioredoxin; thus, the exclusive processing of glucose by the glycolytic pathway would produce a decrease in NADPH availability, increase oxidative stress, and possibly cell death [64]. NADPH serves also as a cofactor of NQO1 detoxifier activity, in the synthesis of fatty acids and myelin, in neurotransmitter turnover, and in the maintenance of redox homeostasis [3,17].

G-6-PDH is one of the key enzymes in the PPP and it has been related to increases in NADPH levels related to the improvement in redox homeostasis [65]; moreover, a decrease of this enzyme has been found in neurodegenerative diseases such as PD [66]. Taking in to account the relevance of this enzyme, we decided to study the level of G-6-PDH in neuronal cultures after oxidative damage induced by VitK3 in the presence or absence of FP. The results in Figure 5 show a decrease in the activity of G-6-PDH after incubation with VitK3. This decrease was abolished when FP was included in the incubation media, producing a great increase in G-6-PDH activity (threefold the control value). This marked increase could be due to the stimulation in synthesis and translocation of Nrf2, an inductor of PPP key enzyme synthesis, triggered by FP after the damage induced with VitK3, as demonstrated previously by our group [17]. Moreover, other authors have previously established the relation between Nrf2 synthesis and translocation, and the expression of PPP key enzymes [67,68].

### 3.4. Influence of S1P Receptors

Classically, FP is thought to develop its effects through the interaction with S1P receptors. We can rule out the possibility of a direct FP–VitK3 pharmacologic interaction, as these drugs do not share or overlap on mechanism of action with either of the enzymes involved in its metabolization, and none of them have any known effect as inductors or inhibitors of metabolic enzymes [69]. We demonstrated in this study the presence of S1P1 receptors in soma and axons of SN4741 neuronal cells (Figure 6).

In order to determine the contribution of these receptor in the protective effect of FP, we performed experiments in the presence of the selective S1P1 competitive antagonist W. In the study of the influence of S1P receptors on the enzymes involved in redox balance, we found intriguing results. Whereas the effect on SOD can be attributable to the interaction of FP with S1P1 receptors—in such a way that we saw an increase in SOD by the increase in free radicals triggered by VitK3, which was totally reverted in presence of FP—when the S1P1 receptor was blocked with the antagonist, the beneficial FP effects disappeared, free radicals increased again [17], and SOD values returned to those found in VitK3-treated neurons (Table 1). The influence of S1P1 receptors on GPX was not so clear. In these experiments, FP demonstrated exertion of its action, not only by its interaction with S1P1 receptors but also by other mechanisms, which could include its interaction with other S1P receptor subtypes expressed in neuronal cells, such as S1P3 and S1P5. FP is a selective but not a specific ligand for the S1P1; in fact, it binds with higher affinity (threefold) to the S1P5 receptor (the less expressed in neurons) and with less affinity (fivefold) for the S1P3 [70]. Unfortunately, there are not enough selective compounds to study receptors other than S1P1. Assuming that FP effect was not due to the interaction of FP with specific S1P receptors, we could still be in the presence of non-receptor mechanisms. These would include an increase in expression of Nrf2, heme oxygenase 1 (OH1), and thioredoxin, and also an increase in Nrf2 translocation to the nucleus [17], with these factors, especially Nrf2 and OH1, being important to increasing antioxidant enzymes such as GPX. Something similar occurred with CAT, where no clear results on the influence of S1P1 were obtained. In these experiments, we did not see effects of VitK3 incubation on CAT activity, nor with the addition of FP. Interestingly when the S1P1 receptor was blocked, we found a decrease in the activity of this enzyme without a clear explanation; however, we could argue that the decrease could have been related to the increase in SOD and GPX previously found, which would reduce the H_2_O_2_ and in turn would decrease CAT [71]. In the study of protein nitrosylation, we found a similar behavior to that in SOD. Neurons incubated with VitK3 showed an increase in protein nitrosylation that was reverted in the presence of FP—the blockade of S1P1 receptor abolished FP beneficial effects, increasing nitrosylation again to levels close to those found in VitK3-treated neurons, agreeing with other authors [72]. All the effects of FP on glycolysis were mediated through its interaction with S1P1 receptors, as all of these effects disappeared when the S1P1 antagonist was present, being affected the basal glycolysis, glycolytic capacity, glycolytic reserve, and the capacity to maintain the aldolase activity. This finding was common with other authors who also found the need of interaction of FP with its receptor to develop its glycolytic protective actions [54,60]. In the study of G-6-PDH activity, the effect of FP was also mediated by the interaction with its specific S1P1 receptor, as the increase in G-6-PDH activity was totally abolished when the S1P1 antagonist was present in the incubation media.

## 4. Conclusions

Overall, our data support that in extreme neuronal situations such as in neurodegenerative diseases, where damaged mitochondria are unable to cover the ATP needs and the oxidative environment turns threatening for the cells, other sources of ATP and antioxidants need to be used. Our results show, on one hand, a beneficial effect of FP, increasing glycolytic and pentose phosphate metabolism and hence ATP production and redox buffering capacity, which are important factors against neurodegeneration. On the other hand, FP also exerts a beneficial effect on mitochondrial oxidative damage, which would improve neuronal condition by improving mitochondrial function and restoring oxidative balance in neurons. Both actions would make this drug a potential therapeutic tool for the treatment of neurodegenerative processes, either to slow progression or alleviate symptoms.

## Figures and Tables

**Figure 1 cells-10-00034-f001:**
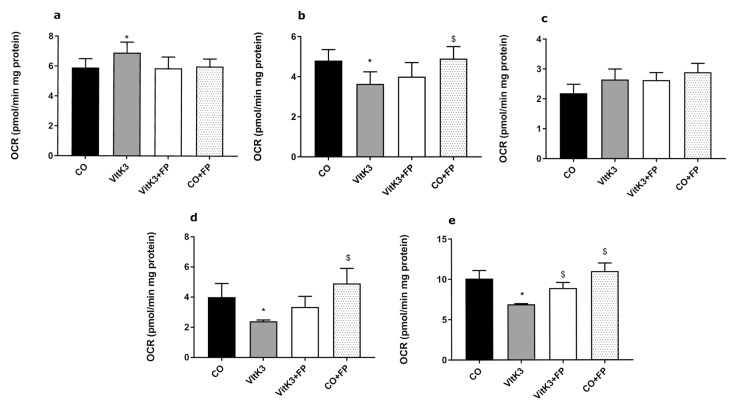
Study of neuronal mitochondrial function after treatment of a cell with menadione (VitK3) in the absence or presence of fingolimod phosphate (FP). The oxygen consumption rate was assessed in (**a**) basal after 2 h of incubation with VitK3; (**b**) ATP production, measured after addition of oligomycin; (**c**) proton leak, considered as the residual oxygen consumption after oligomycin addition; (**d**) spare respiratory capacity (SRC), as the difference between maximal respiration and basal respiration; and (**e**) maximal respiration, obtained as the oxygen consumption after subsequent addition of oligomycin and carbonyl cyanide-4-(trifluoromethoxy)phenylhydrazone (FCCP). The sequence of mitochondrial toxins added was oligomycin, 1 µM; FCCP, 0.5 µM; rotenone/antimycin A, 0.5/0.5 µM. The inclusion of rotenone/antimycin A served to measure respiration by non-mitochondrial processes as these compounds abolish mitochondrial respiration, and it was subtracted in all oxygen consumption rate (OCR) values obtained. Values represent mean and SD of at least five experiments per situation performed in triplicate. (* *p* < 0.05 versus CO, ^$^
*p* < 0.05 versus VitK3).

**Figure 2 cells-10-00034-f002:**
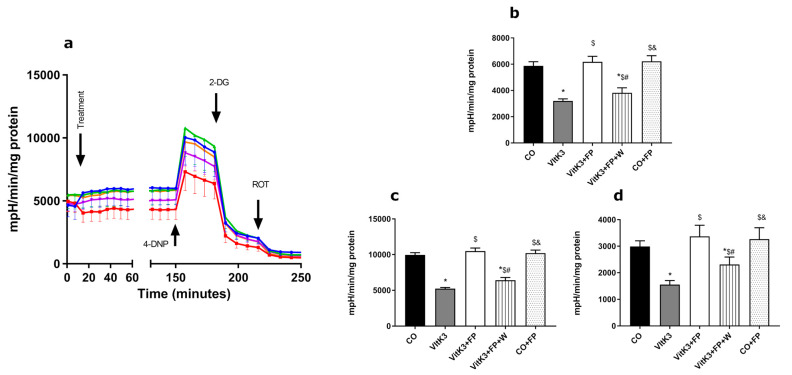
FP recovered the glycolytic damage induced by VitK3 on neuronal SN4741 cells. (**a**) Time course of extracellular acidification rate (ECAR) and glycolytic function after incubation with VitK3 in the presence or absence of FP (blue line: CO; red line: VitK3; green line: VitK3+FP; purple line: VitK3+FP+W). To uncouple mitochondrial respiration from ATP synthesis, we used 2,4-dinitrophenol (2,4-DNP) 100 µM; 2-deoxy-D-glucose (2-DG) 100 mM was used to inhibit glycolysis and rotenone 1 µM was used to inhibit NADH hydrogenase/complex I. (**b**) Basal glycolysis after 2 h of treatment with VitK3 in the presence or absence of FP. (**c**) Glycolytic capacity obtained after incubation with 2,4-DNP and rotenone. (**d**) Glycolytic reserve obtained as the difference between the maximal glycolytic capacity and basal. In all situations, the non-glycolytic ECAR, obtained after addition of rotenone, was subtracted. Values represent mean and SD of at least five experiments per situation performed in triplicate. (* *p* < 0.05 versus CO, $ *p* < 0.05 versus VitK3, # *p* < 0.05 versus VitK3+FP, & *p* < 0.05 versus VitK3+FP+W).

**Figure 3 cells-10-00034-f003:**
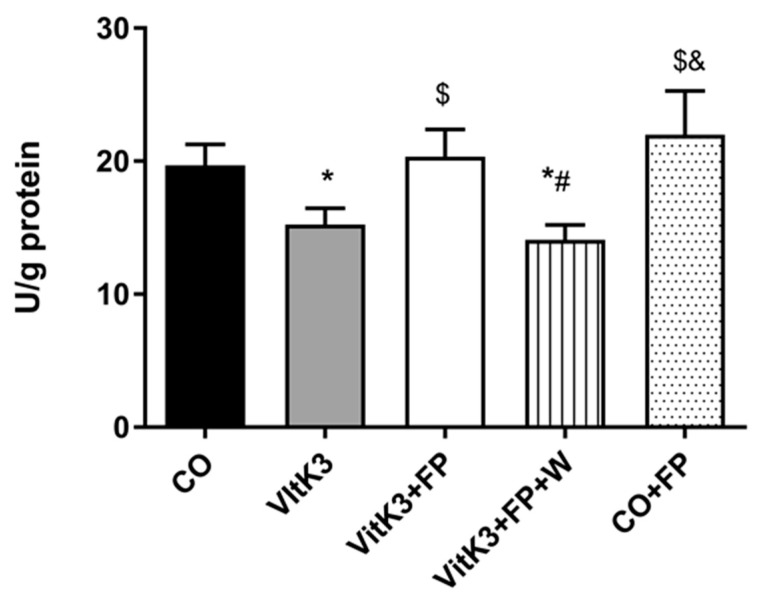
FP recovered the enzyme aldolase from the inhibition induced by VitK3. Aldolase was measured in a homogenate of neurons after 2 hours of incubation with VitK3 in the presence or absence of FP. Values represent mean and SD of at least five experiments per situation performed in triplicate. (* *p* < 0.05 versus CO, $ *p* < 0.05 versus VitK3, # *p* < 0.05 versus VitK3+FP, & *p* < 0.05 versus VitK3+FP+W).

**Figure 4 cells-10-00034-f004:**
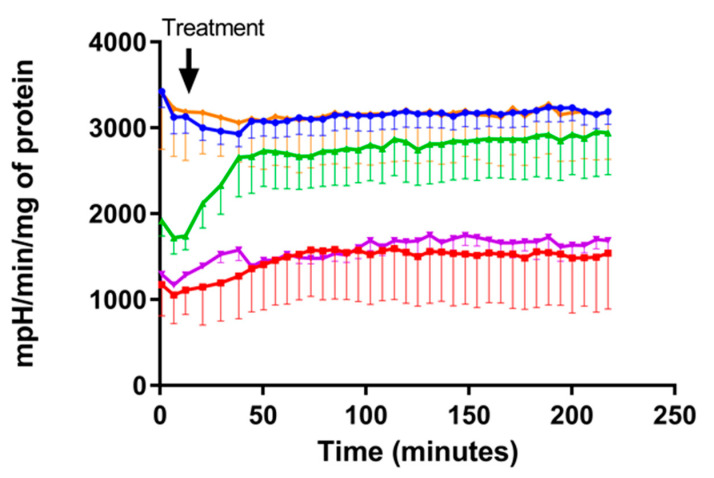
FP was necessary in the medium to maintain its actions. Time course of ECAR after 2 h of incubation with VitK3 in the presence or absence of FP. After 2 h of incubation, VitK3 was removed from the media and cells were incubated for 4 additional hours in the following conditions: two groups were incubated with media (VitK3 group (red line) and CO group (blue line)), one with FP (VitK3+FP group (green line)), one with FP+W (VitK3+FP+W group (purple line)), and one with FP (FP group (orange line). Values represent mean and SD of at least four experiments per situation performed in triplicate.

**Figure 5 cells-10-00034-f005:**
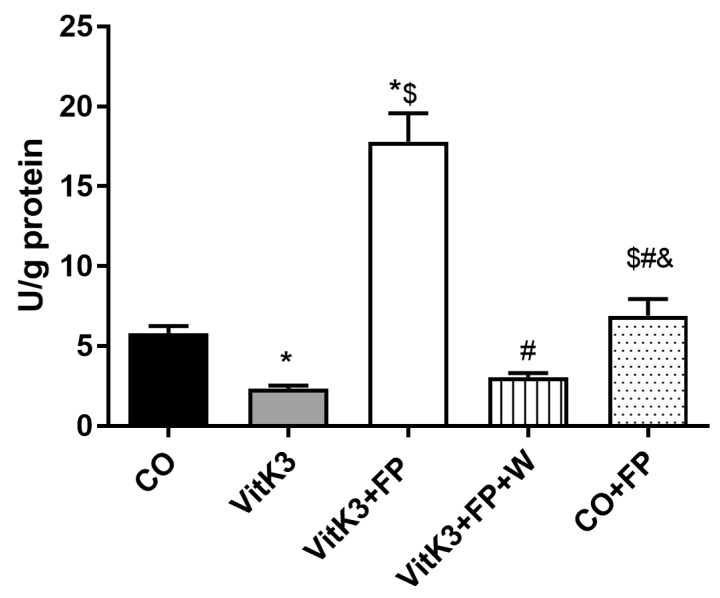
FP increased the enzyme glucose-6-phosphate-dehydrogenase (G-6-PDH) reduced by VitK3. G-6-PDH was measured in a homogenate of neurons after 2 hours of incubation with VitK3 in the presence or absence of FP. Values represent mean and SD of at least five experiments per situation performed in triplicate. (* *p* < 0.05 versus CO, $ *p* < 0.05 versus VitK3, # *p* < 0.05 versus VitK3+FP, & *p* < 0.05 versus VitK3+FP+W).

**Figure 6 cells-10-00034-f006:**
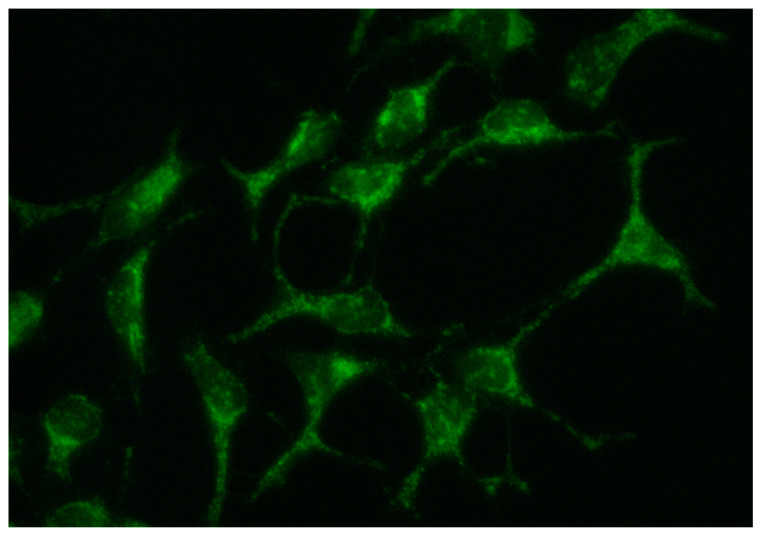
Immunocytochemistry of S1P1 receptor in SN4741 neuronal cells. Image were acquired using an Olympus BX51 epifluorescence microscope at 40× magnification.

**Table 1 cells-10-00034-t001:** Intracellular redox measurements.

	SOD2 U/10^6^ cells	GPX U/10^6^ cells	CAT KU/10^6^ cells	Nit-Prot pmol/10^6^ cells	SOD2/GPX	SOD2/CAT
**CO**	43.1 ± 3.0	21.9 ± 4.6	27.9 ± 3.1	0.76 ± 0.17	2	0.15
**VitK3**	76.4 ± 4.0 (a)	4.5 ± 0.8 (a)	27.1 ± 6.3	4.5 ± 0.35 (a)	17 (a)	0.28
**VitK3+FP**	40.3 ± 3.0 (b)	10.6 ± 0.9 (a)(b)	27.8 ± 4.2	0.45 ± 0.15 (b)	5 (a)(b)	0.14
**VitK3+FP+W**	79.9 ± 6.0 (a)	27.6 ± 4.8 (b)(c)	14.4 ± 2.5 (a)(b)(c)	3.75 ± 0.20 (a)(c)	3 (b)(c)	0.55 (a)(b)(c)
**FP**	50.1 ± 3.0 (b)(c)(d)	23.9 ± 4.0 (b)(c)	44.1 ± 8.5 (a)(b)(c)(d)	1.87 ± 0.5 (a)(b)(c)(d)	2 (b)(c)	0.11 (b)(d)

The main enzymes related to superoxide radical metabolism were measured; also, nitrosylated proteins were measured as a marker of oxidative damage. Each experiment was performed in triplicate; samples were a homogenate from a pool of neuronal cells. The results are expressed as mean ± SD. (a: *p* < 0.05 versus CO, b: *p* < 0.05 versus VitK3, c: *p* < 0.05 versus VitK3+FP, d: *p* < 0.05 versus VitK3+FP+W).

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
