# Peer review of "Neuronal Metabolism and Neuroprotection: Neuroprotective Effect of Fingolimod on Menadione-Induced Mitochondrial Damage"

_cells, 2020, doi:10.3390/cells10010034_

Round 1
Reviewer 1 Report
The paper outlines the potential protective effects of fingolimod on a vitamin K3 induced neuronal function.
I have some questions/comments for the authors that I would like them to address:
- Does vitamin K3 induced neuronal dysfunction represent any neurological disease or is it a purely artificial model of neuronal dysfunction?
- It would be good if the authors could explain why they choose 50nm of fingolimod phosphate (FP) for their experiments and why only one period of incubation was used.
- Did the authors ensure that FP dis interfere with any of the assay employed in the study.
- The authors seem to emphasize the possibility of induction of NO synthase activity but don`t actually measure any actual evidence of this or indeed, whether NO levels are raised. In view of the suggestion of K3 induced oxidative stress it may have been more judicious to assess evidence of reactive oxygen species generation/amelioration following FP treatment.
- Was there any evidence that FP was actually taken into the cells or could the results FP on the plasma membrane interacting with the directly with reagents in the assay?
- Did the authors assess the effect of Vitamin K3 and subsequent FP treatment on the mitochondrial content of the neurons as this may explain the SOD2 results. The mitochondrial dysfunction induced by K3 resulting in an increase in mitochondrial enrichment of the cells in a bid to compensate for the energy impairment being restored to normal levels following FP treatment.
- More explanation is required to explain the GPX results which following K3 and FP treatment.
- I was surprised that the level of the antioxidant, reduced glutathione wasn't assessed in the study as any assessment of the pentose phosphate pathway in neurons without determining glutathione levels seems a bit unusual. How was the flux of the pentose phosphate pathway determined or whether the deficit in G6Pdehydrogenase would actually affect the functioning of the pathway.
- I found the discussion on the glycolysis a bit difficult to follow. Was the glycolytic pathway impaired by Vitamin K3 and restored by FP? Was any assessment of lactate levels as an indicator of glycolytic flux in the face of Mitochondrial dysfunction.
- I had a look at the Vitamin K3 paper and was unable to find out whether K3 actually impaired the mitochondrial respiratory chain or indeed if the activity of the enzymes had been directly assessed. How does K3 impair the mitochondrial and how do the authors suggest FP ameliorates this impairment? I didn`t see a mechanism in the paper.
- Why did the authors assess aldolase activity? Why not phosphofructokinase activity, the rate limiting enzyme glycolysis? Aldolase activity would have little bearing on glycolytic rate.
- It would be good if the authors provided figure to show the impairment induced by K3 and then show the actual effect of FP on this impairment.
- The discussion/results needs to be divided up into sections with subheadings as it lacks clarity at the moment.
Author Response
"Please see the attachment."

Reviewer 2 Report
This study test whether fingolimod, an anti-multiple sclerosis immunomodulatory drug that acts on the sphingosine-1-phosphate (S1P) receptor can provide a neuroprotective effect on menadione (vitamin K3)-induced mitochondrial damage in a neuronal SN4741 cell line. The authors used not a common testing system involving menadione to induce mitochondrial damage. Using predominantly biochemical assays such as nitrotyrosine ELISA, Aldolase assay kit, they showed menadione (vitamin K3) increases oxygen consumption rate, reduce ATP production, etc, which was reverse with fingolimod. Using a competitive S1P receptor inhibitor, they could block the fingolimod effect suggesting the neuroprotective effect was via S1P receptor. Overall, studying mitochondria in neuroprotection is important, but there are several major concerns with this manuscript:
Major concerns:
1- The story of fingolimod inducing neuroprotection is not novel (https://jneuroinflammation.biomedcentral.com/articles/10.1186/s12974-015-0393-6) and (https://www.ncbi.nlm.nih.gov/pmc/articles/PMC5725524/). Also, for the effect to be via S1P receptor is not surprising since is the main receptor for fingolimod.
2-Menadione (vitamin K3) trigger cell death via ROS. However, since fingolimod was directly added with menadione at the same time, it may be a pharmacological interaction of fingolimod on menadione rather than a biological interaction to reduce the oxidative stress. Furthermore, this model is irrelevant to the pathophysiology given that patients often express the disease and have signs of neurodegeneration before therapy. Therefore, I suggest the authors should delay adding fingolimod an hour or so after adding menadione to test that it is capable of reversing the menadione effect.
3-There were many errors in the form of grammar, spelling and others. For example, ‘stablished’, ‘ways’, ‘dipper’, ‘triplicated’, ‘cold’, ‘fails’, ‘G-6-PHD’, ‘whit’, ‘global’. ‘aggressive’. I would have expected the senior authors to have checked the manuscript thoroughly before sending it for review. The y axis in figure 1 is the same for all graphs, but different conditions were measured.
4-The format of the manuscript could be improved. The abstract has a very long introduction and a short result section. It should have been the reverse. The authors keep swapping from menadione to vitamin k3/VitK3 making it confusing to read. Some parts of the manuscript were confusing. For example, what was the relevance of section lines 64-70? It's not related to either fingolimod or the cause of oxidative damage from menadione. They should have discussed more menadione-induced oxidative damage as it is not a regularly used cell insult method in the introduction. Lines 146-153 section should be in the introduction as it is neither a result nor discussion. What evidence to suggest TSPO is found in all cell types?
Minor concerns:
1-provide full names of the abbreviations used in the figure and table legends
2-what is Nrf2 and its role? Discuss
3-mpH/min/mg of what? mpH not pH in panels b-d?
4-define OH1
5- use past tense, so avoid ‘is’ and use ‘was’.
Author Response
Major concerns:
The story of fingolimod inducing neuroprotection is not novel (https://jneuroinflammation.biomedcentral.com/articles/10.1186/s12974-015-0393-6) and (https://www.ncbi.nlm.nih.gov/pmc/articles/PMC5725524/). Also, for the effect to be via S1P receptor is not surprising since is the main receptor for fingolimod.
Response 1. As the referee mention, there are some reports about the neuroprotective effect of FP but in all cases, this effect has been based in an anti-inflammatory effect demonstrated in some papers. In our work we have focussed on the study of the influence of FP on the oxidative and metabolic balance in cultured neurons after damage with the mitochondrial toxic VitK3 being this a new approach to the neuroprotective mechanism of FP. We have consciously focussed on mitochondria as it is the major source of ROS when mitochondrial metabolism fails, being this one of the common events in many neurodegenerative diseases, grouped in global as mitochondriopathies.
The effects seen after incubation with FP, are mostly due to its interaction with the S1P1 receptor, but not all the effects can be attributed only to this interaction, as we have found in our work when blocking the S1P1 receptor with the selective competitive antagonist W123. FP is a selective but not a specific ligand for the S1P1, in fact, it binds with higher affinity (3-fold) to the S1P5 receptor, and almost the same affinity for the S1P4 receptor (Brait V et al. Stroke 47:3053-3056.2016). This point has been clarified in the discussion section.
2-Menadione (vitamin K3) trigger cell death via ROS. However, since fingolimod was directly added with menadione at the same time, it may be pharmacological interaction of fingolimod on menadione rather than a biological interaction to reduce the oxidative stress. Furthermore, this model is irrelevant to the pathophysiology given that patients often express the disease and have signs of neurodegeneration before therapy. Therefore, I suggest the authors should delay adding fingolimod an hour or so after adding menadione to test that it is capable of reversing the menadione effect.
Response 2. It is difficult to imagine a direct or indirect pharmacological interaction, as these drugs do not share or overlap in mechanism of action neither with the enzymes involved in its metabolization and none of both have known effect as inductor or inhibitor of metabolic enzymes (Olivier J D et al. Clin Pharmacokinet. 51:15-28.2012). This point has been clarified in the body of the manuscript.
About the irrelevancy of the model to the pathophysiology of neurodegenerative diseases, we slightly disagree with the referee on this point; as example, FP is giving excellent results in the treatment of multiple sclerosis where neurodegeneration appears at the latest stages of the disease. Demonstrating that a given drug has neuroprotective effects on top of others, like being immunosuppressive or anti-inflammatory, it is and added value of capital importance, especially in diseases which are not originally neurodegeneratives but became at the latest stages of the disease. It could be also useful in neurodegenerative diseases in which a genetic component is present, as the treatment with the drugs could star before the first neurodegenerative events appears (Garcia JC & Bustos RH. Brain Sci. 8:222.218). Furthermore, adding FP after the induction of damage with witK3 we would study the recovery and not the prevention of damage as we intended.
3-There were many errors in the form of grammar, spelling and others. For example, ‘stablished’, ‘ways’, ‘dipper’, ‘triplicated’, ‘cold’, ‘fails’, ‘G-6-PHD’, ‘whit’, ‘global’. ‘aggressive’. I would have expected the senior authors to have checked the manuscript thoroughly before sending it for review. The y axis in figure 1 is the same for all graphs, but different conditions were measured.
Response 3. This first point raised by the referee has been taken into account and corrected in the body of the manuscript.
The y axis in figure 1 is the same because we were measuring the oxygen consumption rate (OCR) and the parameter measured and its units are the same in all conditions, the only change among the different inserts is the scale that has been adapted to a better differentiation of the bars in each condition.
4-The format of the manuscript could be improved. The abstract has a very long introduction and a short result section. It should have been the reverse. The authors keep swapping from menadione to vitamin k3/VitK3 making it confusing to read. Some parts of the manuscript were confusing. For example, what was the relevance of section lines 64-70? It's not related to either fingolimod or the cause of oxidative damage from menadione. They should have discussed more menadione-induced oxidative damage as it is not a regularly used cell insult method in the introduction. Lines 146-153 section should be in the introduction as it is neither a result nor discussion. What evidence to suggest TSPO is found in all cell types?
Response 4. The following points raised by the referee have been taken into account and corrected in the manuscript.
The abstract has been modified following the referee suggestion.
A uniform nomenclature for VitK3 has been followed in the revised version of the manuscript.
Following the referee suggestion, lines 66-70 of the old manuscript, have been removed in the revised version.
As suggested by the referee, the damage induced by VitK3 has been more widely explained in the introduction and lines 146-153 of the old manuscript have been moved to introduction section.
TSPO is a mitochondrial membrane protein associated with the mitochondrial permeability pore and Varga B et al. showed that neuronal cells of embryonic origin expressed this protein. Considering that SN4771 cells are of embryonic origin we could assume that these cell line express this protein (Varga B et al. Neurosci Lett 462: 257-262. 2009). This point has been clarified in the discussion section.
Minor concerns:
1-provide full names of the abbreviations used in the figure and table legends
2-what is Nrf2 and its role? Discuss
3-mpH/min/mg of what? mpH not pH in panels b-d?
4-define OH1
5- use past tense, so avoid ‘is’ and use ‘was’.
Response 5.These points raised by the referee has been taken into account and corrected in the manuscript.

Round 2
Reviewer 1 Report
The authors have addressed all my question adequately and amended the manuscript accordingly. I therefore feel that this paper should now be considered for publication.
Reviewer 2 Report
All comments fully addressed. No further comment.